# The association between parental resilience and emotional/behavioural problems in children with autism spectrum disorders: The mediating role of parenting style

Haiyan Wang[1], Jing Zhao[2]*

1 School of Psychology, Northwest Normal University, Lanzhou, Gansu, China, 2 School of Primary Education, Longnan Normal University, Longnan, Gansu, China

☯ These authors contributed equally to this work.
* zhaojinghope@126.com

## Abstract

Parental psychological resilience plays a crucial role in addressing children's emotional and behavioral problems. However, the association between parental psychological resilience and emotional/behavioral problems of children with Autism Spectrum Disorders (ASD) has been less explored. This study surveyed 258 parents of children with ASD(aged 3–18) who were receiving training at rehabilitation institutions in Shandong Province, China, using questionnaires. Data were analyzed using structural equation modeling to examine the association between parental psychological resilience and emotional/behavioral problems in children with ASD and to identify the underlying pathways. The results indicated that parental psychological resilience is associated with increased prosocial behavior in children with ASD through increased authoritative parenting, while simultaneously being associated with fewer emotional/behavioral problems by reducing permissive and authoritarian parenting styles. This study provides empirical support for family-focused ASD interventions and adds to the growing body of evidence on their effectiveness.

## 1. Introduction

Autism Spectrum Disorders (ASD) is neurodevelopmental disorder characterized by deficits in social communication, restricted interests and repetitive behaviors [1]. The prevalence of ASD in China between 2017 and 2023 was 7 per 1,000, indicating an increasing trend compared to the rate reported before 2017 (26.50 per 10,000) [2]. Emotional and behavioral problems are highly prevalent among children with ASD [3]. These problems can create a persistent cycle of impaired social interaction and emotional dysregulation, adversely affecting individuals across the lifespan [4]. Addressing emotional and behavioral problems in children with ASD requires caregivers' sustained involvement [5,6]. However, most current intervention efforts for

**Data availability statement:** All relevant data are within the paper and its Supporting Information files.

**Funding:** This study was funded by the National Social Science Fund of China (No. BBA1210041)(The funders had no role in study design, data collection and analysis, decision to publish, or preparation of the manuscript.) and the Gansu Province "Innovation Star" Program (No. 2023CXZX-234), with Dr. Haiyan Wang (first author) serving as the principal investigator.

**Competing interests:** The authors have declared that no competing interests exist.

children with ASD often emphasize child behavior modification within clinical or educational contexts, while underutilizing parental psychological resources for dealing with emotional and behavioral problems [7]. Consequently, there is a need to identify and strengthen the internal psychological resources that enable parents to cope effectively. Psychological resilience has been identified by contemporary research as a critical psychological resource [8].

Psychological resilience is the ability to adapt and recover in the face of adversity [9]. Resilience serves as a key protective factor for parents of children with autism. By lowering parenting stress and increasing perceived social support, it indirectly encourages them to adopt more effective parenting practices, such as becoming better able to notice and respond to their autistic child's needs [10]. Research in China indicates that parents of autistic children with higher psychological resilience are better able to positively cope with the multiple challenges in the parenting process, maintain good mental health, and demonstrate more positive parenting functions [11]. Thus, even though the emotional and behavioral problems of children with autism place heavy demands on their parents [12,13], parental psychological resilience can still motivate them to adopt more positive and effective strategies when facing parenting challenges. Highly resilient parents not only experience fewer emotional difficulties themselves but are also better able to create a home atmosphere rich in support and emotional responsiveness for their children [14].

Parenting style is regarded as the most central "proximal process variable" within the family environment [15], it integrates the stable attitudes, behavioral tendencies, and emotional climate that parents display during child-rearing, and is usually classified into three categories: authoritative, authoritarian, and permissive [16]. Parents with higher levels of psychological resilience are more inclined to adopt an authoritative parenting style [17,18], and are less likely to exhibit authoritarian or neglectful tendencies [19]. Authoritative parenting style, characterized by high responsiveness and high demands, can effectively reduce internalizing and externalizing problem behaviors in typically developing children [20] and exert a positive influence on their prosocial behavior [21]. On the contrary, authoritarian or permissive parenting styles may exacerbate children's aggression, anxiety, and depression risks due to a lack of positive modeling and emotional support [16,22]. Thus, it can be inferred that parental psychological resilience is positively related to child development through the mediating pathway of parenting styles, thereby linking to reduced children's emotional and behavioral problems. The specific hypotheses are as follows:

H1. Parental psychological resilience negatively predicts emotional and behavioral problems in children with autism;

H2. Parenting style mediates the relationship between parental psychological resilience and these child outcomes. The hypothesized conceptual model is presented in Fig 1.

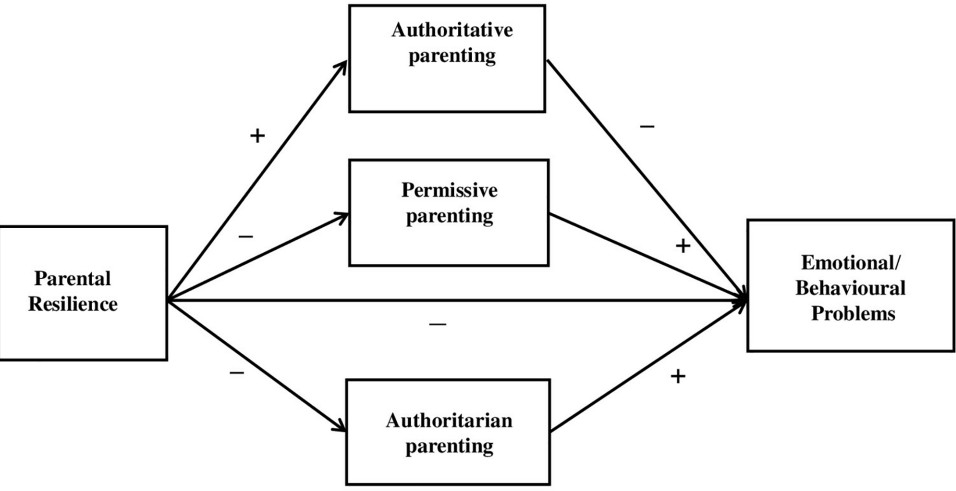

**Fig 1. Assumptions model.**

## 2. Method

### 2.1. Participants and procedure

The participants were recruited from autism rehabilitation institutions in Shandong Province, China. The research team initially distributed 264 questionnaires. After excluding 6 questionnaires where the children's age was ≥ 18 years and a small number of questionnaires with incomplete responses due to privacy concerns, 258 valid questionnaires were ultimately retained. The final dataset used for analysis contained no missing values. Among the final parent sample, 48 (18.6%) were male and 210 (81.4%) were female. Age distribution was as follows: 6 participants (2.3%) were 20–30 years old, 127 (49.2%) were 30–40 years old, and 125 (48.6%) were over 40 years old. Educational background breakdown: 23 participants (8.5%) had completed junior high or below, 53 (20.5%) senior high, 164 (63.6%) university, and 18 (7.0%) postgraduate studies.

Among the fathers, 6 (2.3%) were 20–30 years old, 113 (43.8%) were 30–40, and 139 (53.9%) were over 40. Their educational attainment was distributed as follows: 22 (8.5%) had completed junior high or below, 49 (19.0%) senior high, 161 (62.4%) university, and 26 (10.1%) held postgraduate degrees. Among mothers, 6 (2.3%) were aged 20–30, 131 (50.8%) were 30–40, and 121 (46.9%) were over 40. Educationally, 23 (8.9%) had finished junior high or below, 54 (20.9%) senior high, 162 (62.8%) university, and 19 (7.4%) postgraduate studies.

All children with ASD were diagnosed by child psychiatrists from tertiary hospitals with extensive experience in autism diagnosis, following the diagnostic criteria outlined in the Diagnostic and Statistical Manual of Mental Disorders, Fourth or Fifth Edition (DSM-IV/DSM-5). Among the children, 81 (31.4%) were clinically assessed as having mild ASD (including those formerly diagnosed with Asperger's syndrome), 127 (49.2%) as moderate, and 50 (19.4%) as severe. The children ranged in age from 3 to 18 years ($M = 10.16$, $SD = 4.04$), comprising 212 males (82.2%) and 46 females (17.8%). Parents reported no other serious comorbid conditions.

The questionnaires were distributed on-site by uniformly trained psychology graduate students. The survey was conducted from June 17, 2024, to April 24, 2025. All parents voluntarily completed the questionnaires and signed informed consent forms before participation. Upon completion, they received an assessment feedback report which included evaluations of parental psychological resilience, children's emotional and behavioral characteristics, and parenting recommendations. The data were anonymized, containing no personally identifiable information. The study was approved by the Ethics Committee of Northwest Normal University (approval number: 2024167; date of approval: May 16, 2024).

### 2.2. Measures

**2.2.1. Strengths and difficulties questionnaire.** The parent-report version of the Strengths and Difficulties Questionnaire (SDQ) was utilized in the present study [23,24]. The SDQ is a widely used screening tool for assessing the mental health of children aged 3–17 years. It consists of 25 items divided across five subscales: Emotional Symptoms, Hyperactivity, Conduct Problems, Peer Relationship Problems, and Prosocial Behavior.

Each item is rated on a 3-point Likert scale ranging from 0 (Not True) to 2 (Certainly True), with five items (Items 7, 11, 14, 21, and 25) reverse-scored. The Total Difficulties Score is computed by summing the scores from the Emotional Symptoms, Conduct Problems, Hyperactivity, and Peer Relationship Problems subscales, with higher scores indicating more severe emotional and behavioral difficulties. In contrast, the Prosocial Behavior subscale reflects children's strengths, lower scores on this subscale denote greater deficits in prosocial functioning.

In the current sample, the internal consistency (Cronbach's $a$) was .72 for the Total Difficulties Score and .74 for the Prosocial Behavior subscale, indicating acceptable reliability.

**2.2.2. Resilience scale.** Psychological resilience was measured using the Chinese version of the Connor-Davidson Resilience Scale (CD-RISC), adapted by Yu (2007) from The Chinese University of Hong Kong [25]. The CD-RISC consists of 25 items covering three dimensions: Rresilience, Strength, and Optimism. Each item is rated on a 5-point Likert scale ranging from 1 (Never) to 5 (Almost Always), with higher total scores indicating greater levels of psychological resilience. In the present study, the internal consistency was excellent for the total resilience score (Cronbach's $a = .95$), and acceptable to good for the subscales: Optimism ($a = .68$), Strength ($a = .89$), and Toughness ($a = .93$).

**2.2.3. Parenting style.** Parenting styles were assessed using the short-form Parenting Styles and Dimensions Questionnaire (PSDQ) [26], which has demonstrated satisfactory psychometric properties in Chinese populations [27]. The scale comprises 32 items divided into three primary parenting dimensions: Authoritative (15 items), Authoritarian (12 items), and Permissive (5 items). The Authoritative dimension includes three subscales: Induction, Autonomy Granting, and Warmth. The Authoritarian dimension comprises Physical Coercion, Verbal Hostility, and Non-Reasoning.

Participants responded on a 5-point Likert scale ranging from 1 (Never) to 5 (Always), with higher scores reflecting more frequent use of the respective parenting behaviors. In the current sample, internal consistency coefficients were .93 for Authoritative Parenting, .90 for Authoritarian Parenting, and .67 for Permissive Parenting.

### 2.3. Data analysis

This study utilized SPSS 26.0 and Mplus 8.3 for data processing and analysis. Firstly, descriptive statistics and correlation analyses were conducted in SPSS 26.0; Secondly, independent samples t-tests were performed in SPSS 26.0 to examine whether there were significant differences in the variables based on parental gender and child gender; Finally, structural equation modeling was conducted in Mplus 8.3 to explore the relationship between parental psychological resilience and emotional/behavioral problems in children with autism, as well as the mediating role of parenting styles in this relationship. The Bootstrap method was employed to test the significance of the effect sizes.

In this study, the absolute values of skewness for all variables were less than 2, and the absolute values of kurtosis were less than 4. According to the standard proposed by Kline (2016) [28], if the absolute value of the skewness coefficient is less than 3 and the absolute value of the kurtosis coefficient is less than 10, the variables are considered to follow or approximately follow a normal distribution. Thus, the data did not exhibit severe non-normality. The structural equation modeling analysis was conducted using the Maximum Likelihood Estimation(ML). Model goodness-of-fit was evaluated using the chi-square/degrees of freedom ratio ($\chi^2$/df), comparative fit index (CFI), Tucker-Lewis index (TLI), root mean square error of approximation (RMSEA), and standardized root mean square residual (SRMR). Based on a comprehensive comparison of fit indices by researchers [29], the following criteria were adopted in this study for an acceptable model fit: $\chi^2$/df generally less than 5, CFI and TLI greater than 0.90, and RMSEA and SRMR less than 0.08.

This retrospective study utilized survey data, accessed for research purposes on 29 May 2025. At no point during data extraction, analysis, or reporting did the research team have access to personally identifiable information about study participants. All data were analyzed in anonymized form.

## 3. Result

### 3.1. Common method bias

Common method bias was examined using Harman's single-factor test, following the procedure outlined by Zhou and Long (2004) [30]. An exploratory factor analysis was conducted without rotation, and the results indicated that the first unrotated factor accounted for 17.08% of the total variance, which is well below the recommended threshold of 40%. These findings suggest that common method bias was not a serious concern in this study.

### 3.2. Descriptive statistics and correlation analysis

Descriptive statistics and bivariate correlations among the key variables are presented in Table 1. Parental psychological resilience was significantly positively correlated with authoritative parenting ($r=.58$, $p<.001$), and negatively correlated with both authoritarian parenting ($r=-.20$, $p<.001$) and permissive parenting ($r=-.17$, $p<.01$). In addition, parental psychological resilience showed significant negative correlations with emotional symptoms ($r=-.19$, $p<.01$), conduct problems ($r=-.25$, $p<.001$), and peer relationship problems ($r=-.16$, $p<.05$), while exhibiting a positive correlation with prosocial behavior ($r=.17$, $p<.01$).

Authoritative parenting was negatively correlated with emotional symptoms ($r=-.17$, $p=.008$), conduct problems ($r=-.24$, $p<.001$), and peer relationship problems ($r=-.13$, $p<.05$), and positively correlated with prosocial behavior ($r=.22$, $p<.001$). In contrast, authoritarian parenting showed significant positive correlations with emotional symptoms ($r=.20$, $p<.01$) and conduct problems ($r=.31$, $p<.001$). Permissive parenting was positively associated with

**Table 1. Descriptive statistics and correlation analyses.**

| | 1 | 2 | 3 | 4 | 5 | 6 | 7 | 8 | 9 | 10 | 11 | 12 | 13 | 14 | 15 |
|---|---|---|---|---|---|---|---|---|---|---|---|---|---|---|---|
| 1. Mother age | -- | | | | | | | | | | | | | | |
| 2. Father age | .85** | -- | | | | | | | | | | | | | |
| 3. Mother education | −.04 | .94** | -- | | | | | | | | | | | | |
| 4. Father education | −.01 | −.82** | −.81** | -- | | | | | | | | | | | |
| 5. Child_age | .55** | −.01 | −.07 | .05 | -- | | | | | | | | | | |
| 6. Autism severity | .01 | 0 | −.01 | 0 | .10 | -- | | | | | | | | | |
| 7. parental psychological resilience | −.07 | .06 | .12 | 0 | −.1 | −.08 | -- | | | | | | | | |
| 8. Authoritative parenting | −.05 | −.07 | 0 | .05 | −.10 | −.09 | .58** | -- | | | | | | | |
| 9. Authoritarian parenting | .02 | .04 | .03 | −.01 | .09 | .02 | −.20** | −.37** | -- | | | | | | |
| 10. Permissive parenting | .01 | .06 | .04 | −.01 | 0 | .07 | −.17** | −.22** | .59** | -- | | | | | |
| 11. Emotional Symptoms | .13* | .01 | −.02 | 0 | .21** | .08 | −.19** | −.17** | .19** | .29** | -- | | | | |
| 12. Conduct Problems | .09 | .07 | .05 | −.01 | .09 | −.07 | −.25** | −.24** | .31** | .21** | .41** | -- | | | |
| 13. Hyperactivity | −.02 | −.01 | .03 | −.03 | .06 | .15* | −0.1 | −.12 | .09 | .18** | .24** | .15* | -- | | |
| 14. Peer Relationship | .12 | .11 | .09 | .12 | .22** | .21** | −.16* | −.13* | .12 | .19** | .26** | .18** | .33** | -- | |
| 15. Prosocial Behavior | −.02 | .01 | −.06 | 0 | −.02 | −.24** | .17** | .22** | −.05 | −.07 | −.09 | −.13* | −.35** | −.32** | -- |
| *M* | 2.45 | 2.52 | 2.69 | 2.74 | 10.16 | 1.88 | 63.69 | 55.84 | 26.93 | 12.95 | 2.95 | 1.79 | 6.29 | 5.47 | 3.95 |
| *SD* | 0.54 | 0.55 | 0.74 | 0.75 | 4.04 | 0.70 | 16.48 | 9.78 | 7.37 | 3.04 | 2.06 | 1.38 | 2.20 | 1.76 | 2.48 |

N = 258; *p < 0.05, **p < 0.01, ***p < 0.001.

emotional symptoms ($r=.29$, $p<.001$), conduct problems ($r=.21$, $p<.001$), hyperactivity ($r=.18$, $p<.01$), and peer relationship problems ($r=.19$, $p<.01$).

### 3.3. Analysis of differences in variables by gender

The results of the independent samples t-test indicated that, in terms of parental gender, a significant difference was only observed in educational level ($t=2.00$, $p=0.047$), with male parents having a significantly higher educational level than female parents. No significant differences were found in other variables based on parental gender. The descriptive statistical results for all variables are detailed in Table 2.

The results of the independent samples t-test indicated that, in terms of child gender, a significant difference was only observed in age ($t=2.00$, $p=0.048$), with the mean age of boys being significantly higher than that of girls. No significant differences were found in other variables based on child gender. The descriptive statistical results for all variables are detailed in Table 3.

### 3.4. Structural equation modeling

Consistent with the hypothesized model and correlational results, psychological resilience (including its subdimensions), authoritative parenting, and authoritarian parenting were specified as latent variables in the structural equation model. Variables without an underlying factor structure were treated as observed variables.

To obtain a parsimonious model, preliminary analyses were first conducted on all collected demographic variables. Those showing non-significant associations were subsequently removed, while father's education level, mother's education level, child age, and autism severity were retained as control variables. These covariates were selected because previous research indicates that parental socioeconomic status significantly influences parenting styles, while child developmental stage and symptom

**Table 2. Descriptive Statistics and t-Test Results of Variables by Parental Gender (N = 258).**

| Variable | Male(*n* = 48) | Femal(*n* = 210) | *t* |
|---|---|---|---|
| Age | 2.48 ± 0.50 | 2.46 ± 0.55 | −0.25 |
| Education | 2.88 ± 0.64 | 2.64 ± 0.75 | −2.00* |
| Parental psychological resilience | 66.00 ± 15.79 | 63.16 ± 16.62 | −1.08 |
| Authoritative parenting | 55.06 ± 9.35 | 56.01 ± 9.89 | 0.61 |
| Authoritarian parenting | 27.38 ± 7.89 | 26.82 ± 7.26 | −0.47 |
| Permissive parenting | 13.25 ± 2.81 | 12.89 ± 3.09 | −0.75 |

Note: *$p<0.05$, **$p<0.01$, ***$p<0.001$.

**Table 3. Descriptive Statistics and t-Test Results of Variables by Gender of Children with ASD (N = 258).**

| Variable | Male(*n* = 212) | Femal(*n* = 46) | *t* |
|---|---|---|---|
| Age | 10.36 ± 4.18 | 9.26 ± 3.17 | 2.00* |
| Autism severity | 1.85 ± 0.70 | 2.00 ± 0.73 | −1.28 |
| Emotional Symptoms | 2.92 ± 2.07 | 3.07 ± 2.02 | −0.43 |
| Conduct Problems | 1.80 ± 1.42 | 1.76 ± 1.21 | 0.18 |
| Hyperactivity | 6.27 ± 2.18 | 6.37 ± 2.34 | −0.28 |
| Peer Relationship | 5.44 ± 1.76 | 5.61 ± 1.77 | −0.60 |
| Prosocial Behavior | 4.02 ± 2.48 | 3.63 ± 2.49 | 0.97 |

Note: *$p<0.05$, **$p<0.01$, ***$p<0.001$.

severity are key predictors of emotional and behavioral outcomes. In addition to these structural paths, the model specified a residual correlation between permissive and authoritarian parenting styles. Theoretically, this specification was justified because both styles represent maladaptive parenting strategies that often co-occur in high-stress caregiving contexts. Parents overwhelmed by challenging behaviors may vacillate between harsh control (authoritarian) and submission to child demands (permissive) in an effort to manage distress, resulting in a significant positive correlation between these two dimensions.

The model fit indices reached acceptable levels ($\chi^2(115) = 249.217$, $\chi^2/df = 2.17$, SRMR = .067, RMSEA = .067 (90% CI: .056–.079), CFI = .933, TLI = .904). The final model is presented in Fig 2.

### 3.5. Mediation analysis

Bootstrapping analysis with 5,000 resamples showed that several significant indirect effects through parenting styles. First, authoritative parenting significantly mediated the relationship between parental psychological

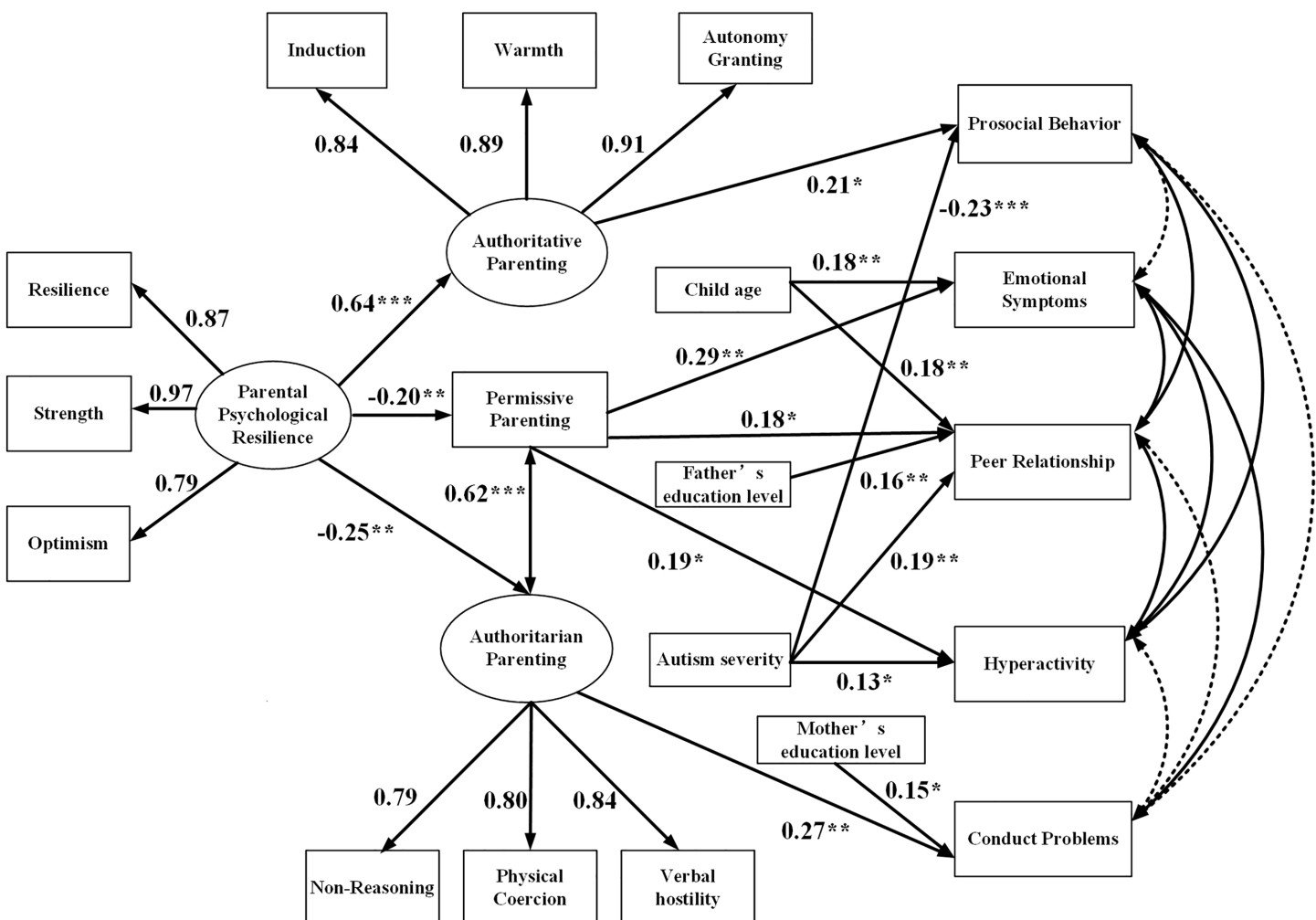

**Fig 2. Structural equation model (N = 258).** *$p < 0.05$, **$p < 0.01$, ***$p < 0.001$, parameter estimates were based on 5,000 bootstrap samples; all coefficients in the figure are standardized estimates. The model specified residual correlation between permissive and authoritarian parenting styles. Parental education levels, child age, and autism severity were included as control variables. For simplicity, only significant paths are labeled. Dashed lines for residual correlations of dependent variables indicate non-significance, while solid lines indicate significance.

resilience and prosocial behavior (indirect effect = .14, p = .025, 95% CI [.017, .256]). Second, permissive parenting significantly mediated the association between resilience and emotional symptoms (indirect effect = −.06, p = .027, 95% CI [−.112, −.007]). Third, authoritarian parenting significantly mediated the relationship between resilience and conduct problems (indirect effect = −.07, p = .015, 95% CI [−.123, −.013]). All confidence intervals excluded zero, indicating that the mediation effects were statistically significant. Descriptive statistics and detailed mediation results are presented in Table 4.

## 4. Discussion

This study examined the relationship between parental psychological resilience and emotional and behavioral problems in children with ASD, with a particular focus on the mediating role of parenting styles. The results revealed that parental psychological resilience was significantly associated with both internalizing problems (e.g., emotional symptoms) and externalizing outcomes (e.g., prosocial behavior and conduct problems) in children with ASD. The results indicate that parental psychological resilience serves as a malleable leverage point for promoting positive development in children with ASD. Future interventions should focus on enhancing parental psychological resilience and strengthening authoritative parenting practices.

Specifically, higher levels of parental psychological resilience were associated with increased prosocial behavior and decreased emotional and behavioral difficulties in children. Importantly, parenting styles fully mediated the relationship between parental psychological resilience and child outcomes. Parents with higher psychological resilience were more likely to adopt an authoritative parenting style, which fostered prosocial behavior in their children. This finding is consistent with previous research [10], indicating that highly resilient parents, when coping with the multiple stressors associated with raising children with autism, are better able to regulate their own emotions and adjust parenting strategies, thereby alleviating both externalizing and internalizing problems in their children. In contrast, lower resilience was associated with greater use of permissive and authoritarian styles, which, in turn, predicted higher levels of emotional symptoms and conduct problems. This findings align with recent person-centered research [20], which found that parenting styles characterized by high authoritative behaviors are associated with fewer internalizing and externalizing problems in children.

Parents of children with ASD who possess higher resilience display greater emotional stability when confronted with their child's dysregulation; they remain composed and deploy effective coping strategies rather than reacting with anxiety or irritability [31]. This within-moment regulation fosters a harmonious context that encourages warmth, structure, and autonomy support, which are key elements of authoritative parenting. [20]. Consequently, resilient parents consistently enact positive practices that advance children's adaptive functioning [32].

Correlational analyses revealed significant negative associations between parental psychological resilience and children's emotional symptoms, conduct problems, and peer relationship difficulties, as well as a significant positive

**Table 4. Bootstrap Test for Significance of Mediation Effects.**

| Mediational Pathway | Effect | SD | Est./S.E. | 95% CI |
|---|---|---|---|---|
| **Parental Psychological Resilience→ authoritative parenting →Prosocial Behavior** | **0.14** | **0.06** | **2.24\*** | [0.017, 0.256] |
| **Parental Psychological Resilience→ permissive parenting→Emotional Symptoms** | **−0.06** | **0.03** | **−2.18\*** | [-0.112, -0.007] |
| **Parental Psychological Resilience→authoritarian parenting →Conduct Problems** | **−0.07** | **0.03** | **−2.44\*** | [-0.123, -0.013] |
| Parental Psychological Resilience→ permissive parenting →Hyperactivity | −0.04 | 0.02 | −1.72 | [-0.080, 0.005] |
| Parental Psychological Resilience→ permissive parenting →Peer Relationship | −0.04 | 0.02 | −1.66 | [-0.080, 0.007] |

Note. *p < 0.05, **p < 0.01, ***p < 0.001; Parameter estimates are based on 5,000 bootstrap resamples. Lower 2.5% and Upper 97.5% indicate the bounds of the 95% bias-corrected confidence interval.

association with prosocial behavior. However, structural equation modeling indicated that, upon inclusion of parenting style as a mediating variable, all direct paths from psychological resilience to emotional and behavioral outcomes became non-significant. This pattern of results suggests a full mediation, indicating that the influence of parental psychological resilience on child outcomes is primarily exerted through parenting behavior. This finding is largely consistent with previous research [33]. The present study further supports the complete mediating role of authoritative parenting style in the relationship between parental psychological resilience and children's psychosocial adaptation. In other words, parental psychological resilience was not directly associated with children's emotional and behavioral outcomes; rather, it functions by prompting them to adopt more positive and supportive parenting approaches, thereby ultimately improving children's psychological adjustment.

These findings underscore the importance of translating internal psychological resources, such as resilience, into observable interactional patterns that directly influence child development. In this context, parenting style functions as the critical conduit through which resilience manifests its effects. As highlighted by Yu et al. (2020) [34], effective ASD symptom management depends on modifying the systems of direct parent-child interaction. Accordingly, the development of scientifically grounded, interaction-focused intervention protocols and caregiver training manuals represents a key strategy for improving behavioral outcomes in children with ASD. Moreover, enhancing the quality and fidelity of these interactional systems is essential to ensure that positive parental psychological resources are effectively translated into measurable improvements in child functioning.

Regarding the magnitude of the effects, the standardized indirect effects of parenting styles observed in this study are considered small-to-medium. However, in the context of complex neurodevelopmental disorders like ASD, these effects have practical significance. Unlike fixed factors such as autism severity or demographics, parenting style is a malleable behavior. Therefore, even modest improvements in parenting practices, driven by enhanced resilience, can translate into meaningful reductions in children's daily behavioral difficulties.

The current study specifically found that resilience was significantly positively associated with authoritative parenting, thereby fostering prosocial behavior in children with ASD. Notably, resilience did not directly predict emotional symptoms or conduct problems. Authoritative parenting, which is marked by a balance of parental warmth, clear expectations, and behavioral guidance [35], typically involves verbal reasoning, empathy, and consistent rule-setting. Such parenting practices not only foster child compliance but also encourage social initiative and cognitive flexibility [36–38]. For children with ASD, authoritative parenting is theoretically aligned with the principles of Applied Behavior Analysis (ABA), the prevailing intervention model that emphasizes positive reinforcement and structured behavior shaping [39]. By offering consistent encouragement, emotional support, and clear behavioral modeling, authoritative parents help scaffold adaptive functioning in children with ASD. The mediating role of authoritative parenting thus provides an empirically supported framework for designing effective, family-based behavioral intervention programs that integrate both psychological and interactional components.

Parents with lower levels of psychological resilience often experience heightened parenting stress, which may predispose them to adopt less adaptive parenting styles such as permissive or authoritarian approaches [40]. The current findings further support this perspective, revealing that parents with low psychological resilience are more likely to engage in permissive parenting, an approach associated with elevated emotional problems in children with Autism Spectrum Disorder (ASD). Permissive parenting is characterized by high responsiveness coupled with low behavioral demands; while parents express warmth and acceptance, they often fail to provide adequate structure or limit-setting [41]. This absence of consistent rules may deprive children with ASD of essential behavioral scaffolding, thereby allowing maladaptive behaviors to persist. At the same time, excessive responsiveness without clear boundaries may unintentionally reinforce dysregulated emotional expressions and excessive dependency behaviors, contributing to further emotional instability.

In addition, low-resilience parents also tend to adopt authoritarian parenting styles, which were associated with increased conduct problems in children with ASD. Authoritarian parenting emphasizes strict control and unquestioning

obedience, marked by high behavioral demands and low emotional responsiveness [42,43]. Children with ASD, who often struggle with rigid or abstract social rules, may find it particularly difficult to interpret and comply with authoritarian expectations. The emotional coldness and lack of affective attunement characteristic of this style may further hinder the child's ability to communicate needs effectively, leading to the manifestation of unmet needs through behavioral dysregulation [44,45].

Interestingly, this study found no association between permissive parenting and conduct problems, nor between authoritarian parenting and emotional symptoms, hyperactivity, or peer problems. These differential patterns may be attributable to the compensatory aspects of each style: the high responsiveness inherent in permissive parenting may partially meet emotional needs, whereas the structured environment of authoritarian parenting may provide a degree of behavioral predictability that supports certain socio-emotional learning processes.

These findings suggest that effective family interaction patterns for children with ASD require a combination of high emotional responsiveness and clearly defined behavioral expectations. Importantly, translating core intervention principles, such as positive reinforcement, modeling, and structured routine, into consistent, day-to-day parenting strategies is essential for increasing the feasibility and impact of home-based interventions. By aligning parenting practices with foundational principles of ASD intervention, caregivers can foster more adaptive developmental outcomes within naturalistic family environments.

Historically, the now-discredited "refrigerator mother" theory, which was prevalent in the 1950s, led to the stigmatization of caregivers by erroneously attributing the cause of autism to cold or emotionally distant parenting [46,47]. Contemporary scientific consensus has since firmly established autism as a neurodevelopmental disorder, with no empirical evidence supporting causal links to parenting style, parental behavior, or emotional neglect [48,49]. Emotional and behavioral difficulties observed in children with Autism Spectrum Disorder (ASD) are now understood as downstream manifestations of underlying neurobiological processes, including atypical brain development and synaptic functioning [50].

Core social communication deficits, typically perceived as a lack of responsiveness or apparent self-absorption, arise from neurobiological differences that impair the processing of social cues and the initiation of reciprocal interaction [51,52]. However, despite advances in neurodevelopmental research, both the etiology and clinical presentation of ASD have historically contributed to a neglect of ecological and systemic perspectives. As a result, the potential ameliorative effects of social interaction, especially within the family context, are frequently overlooked in sociocultural, educational, and clinical discourse [53].

The present study highlights parental psychological resilience as a critical protective factor within the family system that significantly shapes the emotional and behavioral outcomes of children with ASD. These findings underscore the importance of family-based contributions to child development, demonstrating that parental capacities, particularly resilience expressed through adaptive parenting, can meaningfully influence child functioning. An overreliance on professional or institution-based interventions, while undervaluing family-level psychological resources, risks overlooking powerful opportunities for caregiver-mediated developmental support [54,55]. Rebalancing this perspective to recognize and empower families is essential for the design of more holistic and ecologically valid intervention models.

## 5. Research limitations and future directions

This study underscores the vital role of parental psychological resilience and adaptive parenting in addressing emotional and behavioral problems in children with ASD. Nevertheless, several limitations should be noted.

First, the study employed a cross-sectional design, which precludes causal inference. Although mediation analyses offered insights into theoretical pathways, the observed relationships are statistical associations rather than confirmed causal links. Future longitudinal or experimental studies are essential to validate the causal directions of these variables.

Second, reliance on single-informant, self-report measures introduces potential biases. All key variables were reported by parents, which may lead to common method variance (CMV). Although Harman's single-factor test suggested CMV

was not severe, future research should incorporate multi-informant designs (e.g., teacher ratings) and professional assessments (e.g., ADOS-2) to enhance data objectivity. In addition, the internal consistency for the Permissive Parenting subscale was relatively low ($\alpha = .67$). While this is considered acceptable in some psychological research, results involving this specific dimension should be interpreted with caution.

Third, sample characteristics may limit generalizability. Participants were recruited from rehabilitation institutions in a single province in China, which may not fully represent families in other regions or cultural contexts. Furthermore, the gender imbalance (predominantly mothers) and the wide age range of children (3–18 years) may obscure specific paternal perspectives and developmental nuances. Future studies should strive for more diverse, multi-regional samples with balanced gender representation to improve the generalizability and precision of the findings.

## 6. Conclusion

Current research has clearly delineated the positive role of parental psychological resilience in rehabilitation interventions for children with autism, while also underscoring the pivotal mediating role played by family parenting styles in this process. Thus, enhancing parental psychological resilience and guiding them to adopt adaptive parenting strategies represent a critical pathway for improving socio-emotional development outcomes in children with autism. Future efforts to advance child rehabilitation services, if accompanied by simultaneous psychological support and parenting skills training for caregivers, will help establish synergistic mechanisms, thereby enhancing overall intervention outcomes.

## Supporting information

**S1 File. Research survey protocols.**
(DOCX)

**S2 File. Data.**
(XLSX)

## Acknowledgments

The authors want to express their sincere gratitude to all the participants in this study.

## Author contributions

**Data curation:** Haiyan Wang.

**Funding acquisition:** Haiyan Wang.

**Methodology:** Jing Zhao.

**Writing – original draft:** Haiyan Wang, Jing Zhao.

**Writing – review & editing:** Haiyan Wang, Jing Zhao.

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
