## [Decision Letter · Decision Letter 0]

22 Sep 2025

Dear Dr. Zhao,

Thank you for submitting your manuscript to PLOS ONE. After careful consideration, we feel that it has merit but does not fully meet PLOS ONE’s publication criteria as it currently stands. Therefore, we invite you to submit a revised version of the manuscript that addresses the points raised during the review process.

**ACADEMIC EDITOR: I would suggest strongly to rewrite the introduction, and results are not adequate (have suggested some additional analysis) and most of the tests are not presented.**

We look forward to receiving your revised manuscript.

Kind regards,

Samson Nivins, Ph.D

Academic Editor

PLOS ONE

Journal Requirements:

[This study was funded by the National Social Science Fund of China (No. BBA1210041) and the Gansu Province “Innovation Star” project of China (No. 2023CXZX-234).].

4. Please upload a new copy of Figure 1 as the detail is not clear. Please follow the link for more information: https://blogs.plos.org/plos/2019/06/looking-good-tips-for-creating-your-plos-figures-graphics/

Additional Editor Comments:

Editor comments:

Consider changing the title to association. Don’t use effects.

Abstract:

Avoid causative terms, e.g., influence

Add age range of children with ASD in the abstract

Text:

Background: The background is too length, and can you revise and make it concise as there is no concurrent flow when I read and I am losing the true information. Try to keep it to 1.5 pages max.

I would suggest revising the background accordingly. Background about ASD, then introduce about Social emotional problems, then resilience and few theories.

Line 36: remove serious

Add the current prevalence of ASD

There is no bridge between line 56 and below, can authors write a connection or sentence to act as link.

Why do you keep subsection under background – remove. Follow the guidelines.

Check for formatting. Why does author often provide hyphen in the text.

Line 72: Yes, agreed. However, you need to briefly touch base on negative effects of being resilience.

Avoid using the same term ‘Empirical evidence’

Line 107: What do you mean by cognitive neuroscience findings.

I can’t find any research-gap (Why was it placed in hypothesis), and there is no details of what previous people have shown using this.

Methods:

Line 170: Can you make it clear it is parents’ age. If its mother or father, please specify

Line 230: Who were involved in administering and collection details, any authors?

Can you add more details of the aims with stats in data analysis section. For e.g., in results authors are conducting EFA…bivariate analysis and so on…nothing is described.

I would strongly recommend adding all the tests run by the authors to be presented in this section

Whether all the variables are normally distributed? What was the p value for significance.

Provide details of IQ of these children, how many had comorbid

Add details of familial history of mental illness

How were these children diagnosed, who diagnosed and criteria?

Add SEM figure

Results

Line 286: Independent t-test between whom? Nothing is clear. How come 80-20% in sex won’t show any differences. I am bit concerned. I would like to see the results of these data as Tables.

I would strongly recommend considering Sex in the model, or a sub-group analysis, particularly for male.

Can authors look childhood vs adolescence like a sub-group, strongly recommend to do subgroup analysis.

Table 1: Educational attainment of whom – I cant find any details in methods.

Discussion;

Remove subsections

Remove billet points from conclusion

Reviewers' comments:

Reviewer's Responses to Questions

**Comments to the Author**

1. Is the manuscript technically sound, and do the data support the conclusions?

Reviewer #1: Yes

Reviewer #2: Partly

2. Has the statistical analysis been performed appropriately and rigorously?

Reviewer #1: Yes

Reviewer #2: Yes

3. Have the authors made all data underlying the findings in their manuscript fully available?

Reviewer #1: No

Reviewer #2: Yes

4. Is the manuscript presented in an intelligible fashion and written in standard English?

Reviewer #1: Yes

Reviewer #2: Yes

Reviewer #1: In order to improve the quality of the manuscript, certain aspects need to be revised.

1. The second paragraph of the introduction emphasizes the roles of independent and dependent variables, but does not mention mediating variables. A clearer explanation of the relationship between the three is needed.

2. This study investigates parents of children with autism in the context of Chinese culture. Therefore, in the section on “Parental Psychological Resilience,” it is necessary to find some recent studies on the resilience of parents of children with autism in China to better illustrate the value of this study.

3. A model diagram of the hypotheses should be included in the “Research Questions and Hypotheses” section (1.3).

4. Is the sample size for this study appropriate? Is this based on similar studies or statistical conclusions?

5. Please specify whether the parents of children with autism completed the questionnaire voluntarily or in exchange for gifts or compensation.

6. At line 298, determine whether χ2/df < 5.

7. The discussion section should be a key part of the manuscript. Lines 306–314 should discuss the similarities between the results of this study and previous studies.

8. In the limitations section, you can explain the large difference in the number of mothers and fathers in the sample size.

Reviewer #2: The authors' aim was to demonstrate how high parental resilience can improve children's internalizing and externalizing symptoms through the adoption of an authoritative parenting style. The English is fluent and easily understandable. The introduction is well-written and comprehensive. A strengths is the employment of standardized and validated neuropsychological scales.

In my opinion, there are some methodological issues that should be addressed by the authors who request some revisions:

1) METHOD:

The main methodological issues, in my opinion, concern sample selection, inclusion/exclusion criteria, Autism diagnosis, and the assessment of autism level.

My questions are the following:

a) Who made the autism diagnosis? (Please specify whether the diagnosis was made in a level II Center, with a medical and psychological team and whether neuropsychometric tests (ADOS-2, ADI) were instrumental or genetic investigations were performed.)

b) What criteria were used to make the diagnosis? (Please specify if was based on DSM-5 criteria.)

c) Given the extreme heterogeneity of autism (now we consider autism spectrum and phenotypes of autism), was an assessment of the autism level performed? (Please specify whether participants were Level I, II, or II; if this information is not available, it should be included in the study limitations, as the level of autism severity could be a factor influencing parenting stress, resilience, and the presence of internalizing and externalizing problems.)

- What were the exclusion/inclusion criteria? Were neurological and psychiatric comorbidities and relevant clinical conditions excluded? These factors could also influence the above-mentioned aspects.

Minor issues:

- The age range is quite broad. Did the authors perform a data analysis dividing the sample into children and adolescents?

This point could be further explored or included as a weakness.

- The questionnaires were completed by 1/5 of fathers and 4/5 of mothers. How do you interpret this data? Is there a difference in resilience, parenting style, and SDQ scores based on the parent completing the questionnaire?

- Was the parents' social background and educational level analyzed?

- Is there information available indicating whether the parents were receiving parenting support?

2) DISCUSSION:

- The authors state that greater parental resilience predicts better scores in children with ASD on internalizing and externalizing symptoms, however, this relationship may also be bidirectional.

Numerous articles in the literature report higher parental stress in children with ASD and other neurodevelopmental disorders compared to typically developing children, and parental stress may also be linked to child factors such as the severity of autism symptoms, age, and, above all, general adaptive functioning and emotional-behavioral problems (see Operto FF et al. Neuropsychological Profile, Emotional/Behavioral Problems, and Parental Stress in Children with Neurodevelopmental Disorders. Brain Sci. 2021 Apr 30;11(5):584. doi: 10.3390/brainsci11050584; Operto FF et al. Adaptive Behavior, Emotional/Behavioral Problems and Parental Stress in Children With Autism Spectrum Disorder. Front Neurosci. 2021 Nov 25;15:751465. doi: 10.3389/fnins.2021.751465.).

More problematic or more challenging children may place a strain on their parents over time, resulting in increased parenting stress and subsequently lower resilience, with dysfunctional consequences for the entire family well being. Furthermore, we say that parenting stress in chronic situations may increase over time.

This point should be carefully addressed in the discussion.

- Was a standardized assessment of communication and relational symptoms (e.g., ADOS-2), adaptive functioning (VABS, ABAS, etc.), and emotional-behavioral symptoms conducted? This would be very important; if such data are not available, this point should be included in the study limitations.

- The "Limitations and Future Perspectives" section should be expanded to include the aforementioned limitations of the study. For future perspectives, it would be interesting to propose a longitudinal study on evolutionary trajectories over time.

- In my personal opinion, in my conclusions, I would emphasize the point that rehabilitation therapy performed on the child can benefit from a parallel intervention to support the parent (psychological support, parent training) in order to achieve a synergistic effect and improve family well-being.

**Do you want your identity to be public for this peer review?** For information about this choice, including consent withdrawal, please see our Privacy Policy

Reviewer #1: No

Reviewer #2: No

---

## [Author Response · Author response to Decision Letter 1]

3 Nov 2025

Dear reviewers�reviewer comments were submitted as an attachment file.

---

## [Decision Letter · Decision Letter 1]

12 Jan 2026

Dear Dr. Zhao,

Thank you for submitting your manuscript to PLOS ONE. After careful consideration, we feel that it has merit but does not fully meet PLOS ONE’s publication criteria as it currently stands. Therefore, we invite you to submit a revised version of the manuscript that addresses the points raised during the review process.

We look forward to receiving your revised manuscript.

Kind regards,

Samson Nivins, Ph.D

Academic Editor

PLOS One

Journal Requirements:

Additional Editor Comments:

I am the handling editor of your paper. As you know, the revised version submitted by you, should be evaluated by the reviewer who commented earlier. Unfortunately, both the reviewer declined this time.

However, given that major comment was given by me as handling editor, I looked at the reply which you provided, and I am satisfied with it (your response for Editor comments was addressed). Although I invited some additional reviewers to look at it, who has suggested some minor suggestions.

I hope you can address those comments promptly.

Reviewers' comments:

Reviewer's Responses to Questions

**Comments to the Author**

Reviewer #3: (No Response)

2. Is the manuscript technically sound, and do the data support the conclusions?

Reviewer #3: Yes

3. Has the statistical analysis been performed appropriately and rigorously?

Reviewer #3: (No Response)

4. Have the authors made all data underlying the findings in their manuscript fully available?

Reviewer #3: Yes

5. Is the manuscript presented in an intelligible fashion and written in standard English?

Reviewer #3: Yes

Reviewer #3: The Effect of Parental Resilience on Emotional and Behavioural Problems in Children with Autism: The Mediating Role of Parenting Style

This manuscript examines the association between parental psychological resilience and emotional and behavioral problems in children with Autism Spectrum Disorder (ASD), with parenting style as a mediating mechanism. The topic is relevant, the theoretical framework is appropriate, and the use of validated instruments and structural equation modeling (SEM) is generally sound. The study contributes useful evidence to the literature on family-based approaches to ASD intervention.

However, several issues related to study design, interpretation of findings, figure clarity, and reporting transparency should be addressed before the manuscript can be considered for publication.

Major Comments

1. Study design and causal interpretation

The study employs a cross-sectional design, yet causal language is used throughout the manuscript (e.g., “resilience promotes prosocial behavior,” “mitigates emotional symptoms”). While mediation analyses were conducted, such analyses do not establish causal pathways in cross-sectional data.

Recommendation:

Please revise the manuscript to consistently use associational language and clearly state that the mediation findings are statistical rather than causal in nature, particularly in the Abstract, Results, and Discussion sections.

2. Common method bias and single-informant data

All key variables (parental resilience, parenting style, and child outcomes) were reported by the same parent respondent. Although Harman’s single-factor test was conducted, this method alone is limited in its ability to rule out common method variance.

Recommendation:

Explicitly acknowledge this limitation and discuss how shared method variance may have influenced the observed relationships. Future research directions should emphasize multi-informant or observational designs.

3. Sample characteristics and generalizability

Participants were recruited from autism intervention institutions in a single province in China, and the majority of respondents were mothers. These factors may limit the generalizability of the findings to other regions, cultural contexts, caregiver roles, or families without access to intervention services.

Recommendation:

Expand the discussion of generalizability limitations and clearly state the population to which the findings are most applicable.

4. Measurement considerations

The permissive parenting subscale demonstrated relatively low internal consistency (α = .67) compared to other measures used in the study.

Recommendation:

Acknowledge this limitation and briefly discuss its potential impact on mediation results involving permissive parenting.

5. Clarity and quality of Figure 1

Figure 1 (Structural equation model) is difficult to interpret in its current form. The font size, path coefficients, and labels are small and unclear, which makes it challenging to follow the model structure and key findings.

Recommendation:

Please improve the clarity of Figure 1 by:

Increasing resolution and font size

Ensuring all paths and coefficients are clearly legible

Simplifying the figure if possible, or providing a more detailed caption explaining the main paths

Clear and readable figures are essential for accurate interpretation of SEM results.

6. Covariates and model specification

Parental age and child age were included as covariates for selected outcomes; however, the rationale for their inclusion is not fully explained, nor is it clear why other demographic variables (e.g., parental education) were not controlled.

Recommendation:

Please clarify the theoretical or empirical rationale for the selected covariates and explain the decision-making process for including or excluding other demographic variables in the SEM.

7. Missing data handling

The manuscript states that questionnaires with “substantial missing data” were excluded, but no criteria are provided, and the handling of remaining missing data is not described.

Recommendation:

Please clarify:

The criteria used to define “substantial” missing data

Whether any missing values remained in the final dataset

The estimation method used in Mplus for handling missing data (e.g., FIML)

8. Interpretation of effect sizes

The Results and Discussion sections focus primarily on statistical significance, with limited discussion of the magnitude or practical significance of the observed effects.

Recommendation:

Please include a brief discussion of the practical or clinical relevance of the effect sizes, particularly in relation to family-based interventions for children with ASD.

Minor Comments

Terminology consistency:

Terms such as parental psychological resilience, parental resilience, and family resilience are used interchangeably. Please define these constructs clearly and use consistent terminology throughout the manuscript.

Residual correlations:

Provide a clearer justification for correlating residuals between permissive and authoritarian parenting beyond reliance on modification indices.

**Do you want your identity to be public for this peer review?** For information about this choice, including consent withdrawal, please see our Privacy Policy

Reviewer #3: No

---

## [Author Response · Author response to Decision Letter 2]

18 Jan 2026

Response to reviews

Dear Editor,

We now resubmit our manuscript after substantial revision. We thank the reviewers for their excellent comments, which have helped tremendously to improve the quality of the paper. Attached are the point-by-point responses to the reviewer comments and the revised version of our paper.

We hope that the paper is now acceptable for publication. Thanks for considering our paper for publication in your esteemed journal.

Below, reviewer's comments and our responses are presented in black and blue, respectively.

Reviewer

This manuscript examines the association between parental psychological resilience and emotional and behavioral problems in children with Autism Spectrum Disorder (ASD), with parenting style as a mediating mechanism. The topic is relevant, the theoretical framework is appropriate, and the use of validated instruments and structural equation modeling (SEM) is generally sound. The study contributes useful evidence to the literature on family-based approaches to ASD intervention.

However, several issues related to study design, interpretation of findings, figure clarity, and reporting transparency should be addressed before the manuscript can be considered for publication.

Response: We are grateful for your encouraging comments and constructive suggestions. Each and every one has been carefully addressed point by point in our revised manuscript.

Major Comments

1. Study design and causal interpretation

The study employs a cross-sectional design, yet causal language is used throughout the manuscript (e.g., “resilience promotes prosocial behavior,” “mitigates emotional symptoms”). While mediation analyses were conducted, such analyses do not establish causal pathways in cross-sectional data.

Recommendation:

Please revise the manuscript to consistently use associational language and clearly state that the mediation findings are statistical rather than causal in nature, particularly in the Abstract, Results, and Discussion sections.

Response: We sincerely thank the reviewer for this critical observation. We agree that given the cross-sectional nature of our study, causal language was inappropriate. We have carefully revised the manuscript to use associational language throughout the Abstract, Introduction, and Discussion sections. Furthermore, as recommended, we have explicitly stated in the Discussion section that the mediation findings represent statistical associations rather than confirmed causal pathways, and we have emphasized the need for future longitudinal research to validate these relationships.

2. Common method bias and single-informant data

All key variables (parental resilience, parenting style, and child outcomes) were reported by the same parent respondent. Although Harman’s single-factor test was conducted, this method alone is limited in its ability to rule out common method variance.

Recommendation:

Explicitly acknowledge this limitation and discuss how shared method variance may have influenced the observed relationships. Future research directions should emphasize multi-informant or observational designs.

Response: We appreciate the reviewer highlighting this important methodological limitation. We acknowledge that relying exclusively on parent-reported data for both independent and dependent variables introduces the risk of common method variance (CMV), which may inflate the observed relationships, and we agree that Harman’s single-factor test alone is insufficient to completely rule out this bias. In response to your recommendation, we have expanded the "Limitations" section to explicitly discuss the potential influence of shared method variance on our results. Furthermore, we have added a recommendation for future studies to utilize multi-informant designs (e.g., collecting data from teachers or clinicians) and observational measures to enhance the objectivity of the findings.

3. Sample characteristics and generalizability

Participants were recruited from autism intervention institutions in a single province in China, and the majority of respondents were mothers. These factors may limit the generalizability of the findings to other regions, cultural contexts, caregiver roles, or families without access to intervention services.

Recommendation:

Expand the discussion of generalizability limitations and clearly state the population to which the findings are most applicable.

Response: We agree with the reviewer that the recruitment of participants from a single province and the predominance of mothers in the sample limit the generalizability of our findings. In accordance with your recommendation, we have expanded the discussion on limitations to explicitly acknowledge these geographical and demographic constraints. We have clarified that the findings may not be fully generalizable to other cultural contexts or to fathers, and we have emphasized the need for future multi-regional studies with more balanced gender representation.

4. Measurement considerations

The permissive parenting subscale demonstrated relatively low internal consistency (α = .67) compared to other measures used in the study.

Recommendation:

Acknowledge this limitation and briefly discuss its potential impact on mediation results involving permissive parenting.

Response: Thank you for pointing this out. We acknowledge that the Cronbach’s alpha for the Permissive Parenting subscale (.67) was relatively low compared to the other measures. We have added a statement in the "Limitations" section to explicitly recognize this issue. We also advise readers to interpret the mediation results involving permissive parenting with appropriate caution due to this lower internal consistency.

5. Clarity and quality of Figure 1

Figure 1 (Structural equation model) is difficult to interpret in its current form. The font size, path coefficients, and labels are small and unclear, which makes it challenging to follow the model structure and key findings.

Recommendation:

Please improve the clarity of Figure 1 by:

Increasing resolution and font size

Ensuring all paths and coefficients are clearly legible

Simplifying the figure if possible, or providing a more detailed caption explaining the main paths

Clear and readable figures are essential for accurate interpretation of SEM results.

Response:We thank the reviewer for this important feedback regarding the figure's readability.

Clarification on Figure Numbering:We noticed the comment refers to "path coefficients" and the "Structural Equation Model." In our manuscript, Figure 1 is the conceptual Assumptions Model (theoretical framework without coefficients), while Figure 2 displays the SEM results with specific path coefficients. We assume the reviewer intended to critique the clarity of Figure 2 (the Results Model).

Improvements Made:We have completely processed Figure 2 to address the visibility issues:

1.Enlarged Typography: We have significantly increased the font size of all path coefficients and variable labels to ensure they are clearly legible.

2.High-Resolution: The figure has been replaced with a high-resolution version to eliminate blurriness.

3.Visual Clarity regarding Simplification: Regarding the suggestion to simplify: We carefully considered removing the control variables from the figure. However, we decided to retain them to ensure the transparency and completeness of the reported statistical model, allowing readers to see the exact estimated paths. Instead of removing them, we improved the layout and line contrast to minimize visual clutter while keeping the model specification precise.

The revised Figure 2 is now presented in the revised manuscript.

6. Covariates and model specification

Parental age and child age were included as covariates for selected outcomes; however, the rationale for their inclusion is not fully explained, nor is it clear why other demographic variables (e.g., parental education) were not controlled.

Recommendation:

Please clarify the theoretical or empirical rationale for the selected covariates and explain the decision-making process for including or excluding other demographic variables in the SEM.

Response: We appreciate the reviewer’s attention to model specification. We wish to clarify our variable selection process. In our preliminary analysis, we screened all collected demographic variables as potential control variables. To maintain model parsimony, demographic variables that showed no significant associations with the outcome variables were excluded from the final Structural Equation Model. However, we explicitly retained parental education levels, child age, and autism severity as control variables in the final model (as noted in Section 3.4), because they showed significant effects. Regarding the reviewer's concern about "parental education": it was indeed controlled for in the final model. We retained these specific variables because existing literature consistently shows that child developmental stage and symptom severity are fundamental predictors of behavioral problems, while parental education (as a proxy for SES) is a stable predictor of parenting styles. We have added a sentence in Section 3.4 to explicitly state this selection rationale.

7. Missing data handling

The manuscript states that questionnaires with “substantial missing data” were excluded, but no criteria are provided, and the handling of remaining missing data is not described.

Recommendation:

Please clarify:

The criteria used to define “substantial” missing data

Whether any missing values remained in the final dataset

The estimation method used in Mplus for handling missing data (e.g., FIML)

Response: We apologize for the lack of clarity regarding data screening. In this study, questionnaires with incomplete responses (often due to participants declining to answer specific privacy-related items) were strictly excluded from the dataset (listwise deletion) to ensure data quality. Consequently, the final sample of 258 participants contained no missing data. We have revised Section 2.1 to clarify this exclusion criterion and the completeness of the final dataset.

8. Interpretation of effect sizes

The Results and Discussion sections focus primarily on statistical significance, with limited discussion of the magnitude or practical significance of the observed effects.

Recommendation:

Please include a brief discussion of the practical or clinical relevance of the effect sizes, particularly in relation to family-based interventions for children with ASD.

Response: Thank you for this valuable suggestion. We agree that statistical significance should be contextualized with practical relevance. We have added a paragraph in the Discussion section to interpret the effect sizes. We argue that while the standardized coefficients are small-to-medium, they are clinically meaningful because parenting style is a modifiable factor. Identifying such malleable targets is crucial for family-based interventions, where even incremental improvements can significantly enhance the quality of life for families of children with ASD.

Minor Comments

Terminology consistency:

Terms such as parental psychological resilience, parental resilience, and family resilience are used interchangeably. Please define these constructs clearly and use consistent terminology throughout the manuscript.

Response: We apologize for the inconsistent terminology. We have carefully reviewed the manuscript and standardized the terminology. We now use "Parental Psychological Resilience" consistently throughout the text to refer to the construct. Occurrences of other variations have been corrected.

Residual correlations:

Provide a clearer justification for correlating residuals between permissive and authoritarian parenting beyond reliance on modification indices.

Response: Thank you for requesting a clearer justification. We allowed the residuals of permissive and authoritarian parenting to correlate not merely based on modification indices, but on theoretical grounds. Both styles represent maladaptive approaches characterized by a lack of constructive structure or emotional support. Theoretically, this specification was justified because both styles represent maladaptive parenting strategies that often co-occur in high-stress caregiving contexts. According to Coercion Theory [1], parents overwhelmed by challenging behaviors may vacillate between harsh control (authoritarian) and submission to child demands (permissive) in an effort to manage distress, resulting in a significant positive correlation between these two dimensions.We have added this theoretical justification to Section 3.4.

[1]Patterson, G. R. (1982). Coercive family process. Eugene, OR: Castalia.

---

## [Editor Report · Decision Letter 2]

19 Jan 2026

The association between Parental Resilience and Emotional/Behavioural Problems in children with Autism Spectrum Disorders: The mediating role of Parenting Style

PONE-D-25-39253R2

Dear Dr. Zhao,

We’re pleased to inform you that your manuscript has been judged scientifically suitable for publication and will be formally accepted for publication once it meets all outstanding technical requirements.

Kind regards,

Samson Nivins, Ph.D

Academic Editor

PLOS One

Additional Editor Comments (optional):

Thank you. I have checked all the comments raised by the reviewer, and was adequately addressed.
---

## [Editor Report · Acceptance letter]

PONE-D-25-39253R2

PLOS One

Dear Dr. Zhao,

I'm pleased to inform you that your manuscript has been deemed suitable for publication in PLOS One. Congratulations! Your manuscript is now being handed over to our production team.

Kind regards,

on behalf of

Dr. Samson Nivins

Academic Editor

PLOS One